# Lensless Three-Dimensional Imaging under Photon-Starved Conditions

**DOI:** 10.3390/s23042336

**Published:** 2023-02-20

**Authors:** Jae-Young Jang, Myungjin Cho

**Affiliations:** 1Department of Optometry, Eulji University, 553 Sanseong-daero, Sujeong-gu, Seongnam-si 13135, Kyonggi-do, Republic of Korea; 2Research Center for Hyper-Connected Convergence Technology, School of ICT, Robotics, and Mechanical Engineering, Institute of Information and Telecommunication Convergence (IITC), Hankyong National University, 327 Chungang-ro, Anseong 17579, Kyonggi-do, Republic of Korea

**Keywords:** Bayesian estimation, computational photon counting method, diffraction grating imaging, lensless 3D imaging, maximum likelihood estimation, multiple observation photon counting method, three-dimensional imaging

## Abstract

In this paper, we propose a lensless three-dimensional (3D) imaging under photon-starved conditions using diffraction grating and computational photon counting method. In conventional 3D imaging with and without the lens, 3D visualization of objects under photon-starved conditions may be difficult due to lack of photons. To solve this problem, our proposed method uses diffraction grating imaging as lensless 3D imaging and computational photon counting method for 3D visualization of objects under these conditions. In addition, to improve the visual quality of 3D images under severely photon-starved conditions, in this paper, multiple observation photon counting method with advanced statistical estimation such as Bayesian estimation is proposed. Multiple observation photon counting method can estimate the more accurate 3D images by remedying the random errors of photon occurrence because it can increase the samples of photons. To prove the ability of our proposed method, we implement the optical experiments and calculate the peak sidelobe ratio as the performance metric.

## 1. Introduction

There have been various three-dimensional (3D) imaging techniques for acquiring full parallax images from a 3D scene; direct pickup in integral imaging [1,2,3,4], synthetic aperture integral imaging [5,6], light field imaging [7,8], lensless 3D imaging [9,10], and so on. In particular, lensless 3D imaging is free from diffraction and aberrations that occur in lens array-based acquisition methods, and has advantages over camera array or moving camera-based acquisition methods in point of view for cost effectiveness and system complexity. Diffraction grating imaging [11,12,13,14], which is one of lensless 3D imaging, consists of a transmission grating to generate diffraction images and a camera or image sensor to acquire the generated diffraction image array. These diffraction images are used for the computational 3D reconstructions [11,12,13,14]. Various diffraction image acquisition methods for improving the resolution of the reconstructed 3D images have been studied [13,14]. The computational 3D reconstruction method considering the periodic imaging characteristics corresponding to the spatial depth of a diffraction image array is studied [13].

However, since the conventional lensless 3D imaging may not produce 3D images under photon-starved conditions, a new lensless 3D imaging technique is required. Therefore, in this paper, we propose a new lensless 3D imaging under photon-starved conditions which uses the diffraction grating and computational photon counting method [15,16,17,18,19,20,21,22,23,24]. diffraction images captured by the conventional lensless 3D imaging may have the small number of photons under these conditions. Thus, the resolution and the visual quality of 3D images may be degraded. To overcome this problem, photon counting method can be utilized. In this paper, we introduce the computational photon counting method for convenience, since the physical photon counting detector is not cost-effective.

Computational photon counting method uses the random process of Poisson distribution because photons occur rarely in unit time and space [25]. In addition, for estimating the images, advanced statistical estimations such as maximum likelihood estimation (MLE) and Bayesian estimation are used. However, since lensless 3D imaging has low light efficiency and detected photons may decrease, more accurate estimation method is required for estimating the images under photon-starved conditions. Therefore, in this paper, we introduce a new 3D reconstruction of computational photon counting method by using multiple observations of photon counting method, which is called as *N* observation photon counting method. It can generate multiple 2D photon counting images from the single diffraction image recorded by diffraction grating imaging. It means that the number of samples for photons can increase for the accurate estimation. Thus, the reconstructed 3D images with improved visual quality can be obtained by our proposed method.

This paper is organized as follows. In Section 2, we present the basic concept of lensless 3D imaging. Then, we describe the computational photon counting method with advanced statistical estimations in Section 3. Our proposed method is expressed in Section 4. To show the feasibility of our proposed method, we show the experimental results in Section 5. Finally, we make the conclusion with summary in Section 6.

## 2. Lensless Three-Dimensional Imaging and Computational Reconstruction

A diffraction grating imaging is a lensless three-dimensional (3D) imaging technique that can acquire shape and depth information from the 3D scene. Diffraction grating imaging system is generally composed of a diffraction grating and an image acquisition device such as a camera for acquiring a diffraction image array (DIA) [11,12,13,14]. In diffraction grating imaging, light scattered from a 3D object by the light illumination is diffracted by a diffraction grating located on the optical path and formed into the DIA having full parallax. when a light wave is incident perpendicular to the grating with linear gap, the diffracted light has a maximum intensity at the diffraction angle $\theta_{m}$ given by the diffraction equation as $a\cdot \sin\theta_{m}=m\lambda$, where *a* is the gap between adjacent slits, and $\theta_{m}$ is the angle between the diffracted ray and the grating’s normal vector, *m* is the integer representing the diffraction order, and λ is the wavelength of the light source. The DIA is generated as a virtual image at the same depth as the 3D object and can be observed with the naked eye through a transmission diffraction grating or obtained by an acquisition device such as a camera. Considering the diffraction equation and the characteristic that the imaging depth is the same as the depth of the 3D object, it can be inferred that the distance between the diffraction images in the DIA has a constant value corresponding to the depth of the 3D object. These characteristics provide clues that enable reconstruction of the 3D scene from the DIA.

### 2.1. Geometric Relations

Figure 1 illustrates the geometric relationship of the principal variables used in the analysis of diffraction grating imaging. In Figure 1a, the distance between the diffraction grating and the imaging lens, which is the main component of diffraction grating imaging, is separated by *d*. In the coordinate system based on the imaging lens, it is assumed that a monochromatic light is irradiated onto the point object located on the ($x_{o},y_{o},z_{o}$), and the diffraction image of the point object is formed on the diffraction image (DI) plane due to the diffraction grating. At this time, the coordinates of each diffraction image are $DI(x_{mth},y_{nth},z_{o})$, where *m* and *n* are an integer representing the diffraction order in *x* and *y* directions, and the point object is regarded as a 0th order diffraction image. The angle formed by the ±1st order diffraction image and the normal line is θ, which is given by the diffraction formula mentioned above, and the depth of each diffraction image (the distance between the diffraction grating and the diffraction image) is $|z_{o}-d|$. *X* is the spatial period between diffraction images in the DIA and it has the constant value corresponding to the depth of the object. Since the ±1st order diffraction images on the DI plane are the virtual image, the light involved in forming the image on the DI pick-up plane away from the imaging lens by $z_{I}$ comes from the point object. Considering this reason, in Figure 1a, the chief ray starting from the point object is expressed as a solid line, and the virtual ray from the ±1st order diffraction images is expressed as a dotted line. The right side of Figure 1a is an example of the acquired DIA.

Considering the position of the point object and the diffraction order, the *x*-coordinate of the diffraction image $x_{mth}$ is given by
(1)$$x_{mth}=x_{o}+\left|z_{o}-d\right|\tan\left(\sin^{-1}\left(\frac{m\lambda }{a}\right)\right),$$
where *m* is the integer representing the diffraction order and $|z_{o}-d|$ is a distance between the object and the diffraction grating.

From the geometrical relationship and Equation (Equation 1), one-dimensional (1D) form of the spatial period of the DIA depending on the object’s depth can be given by $|x_{(s)th}-x_{(s-1)th}|$, where s=0 or 1. Then, the spatial period, *X*, depending on the object’s depth is given by
(2)$$X_{z_{o}}=\left|z_{o}-d\right|\tan\left(\sin^{-1}\left(\frac{\lambda }{a}\right)\right).$$

Figure 1b shows the geometric relationship of the difference in spatial period for point objects placed at different depths and the obtained DIA. As shown in Figure 1b and Equation (Equation 2), the depth information of the object located in 3D space is recorded as spatial period information in a two-dimensional (2D) DIA.

### 2.2. Imaging Formation of Diffraction Grating Imaging

Imaging formation of DIA in diffraction grating imaging can be explained in terms of intensity impulse response and scaled object intensity. In conventional 2D imaging, image intensity can be represented as $g\left(x_{D}\right)=h\left(x_{D}\right)*f\left(x_{D}\right)$, where ∗ denotes convolution, $x_{D}$ represents the *x*-coordinate on DIA, $h(x_{D})$ represents the intensity impulse response, and $f(x_{D})$ represents the scaled object intensity considering the image magnification.

On the other hand, since the intensity impulse response and scaled object intensity depend on the object’s depth $z_{o}$, the image intensity for 3D object can be rewritten as $g\left(x_{D}\right){|}_{z_{o}}=f\left(x_{D}\right){{|}_{z_{o}}*h\left(x_{D}\right)|}_{z_{o}}$ with $z_{o}$ dependency. Considering the continuously distributed intensity of 3D volume object, the $z_{o}$ dependent image intensity can be given by
(3)$$g\left(x_{D}\right)=\int h\left(z_{o},x_{D}\right)*f\left(z_{o},x_{D}\right)dz_{o}.$$

Here, the intensity impulse response $h(z_{o},x_{D})$ corresponding to the diffraction grating can be approximated by a δ-function array [11]. In addition, considering the characteristic that DIA is formed periodically at the object’s depth $z_{o}$, the intensity impulse response can be given by
(4)$$h\left(z_{o},x_{D}\right)=\sum_{k=-1}^{1}\delta \left(x_{o}-kX_{z_{o}}\right),$$
where $X_{z_{o}}$ can be calculated from Equation (Equation 2) as the spatial period at the object’s depth. Next, we consider a scaled object intensity in Equation (Equation 3). The scaled object intensity can be represented as the object intensity considering the imaging magnification and can be given by
(5)$$f\left(z_{o},x_{D}\right)=\left|\frac{z_{I}}{z_{o}}\right|f_{o}\left(z_{o},-x_{o}\right).$$

Therefore, the intensity of DIA can be derived by substituting Equations (4) and (5) into Equation (Equation 3), and it is given by
(6)$$g\left(x_{D}\right)=\int \int \sum_{k=-1}^{1}\delta \left(x_{o}-kX_{z_{o}}\right)\left|\frac{z_{I}}{z_{o}}\right|f_{o}\left(z_{o},-x_{o}\right)dx_{o}dz_{o}.$$

### 2.3. Computational Reconstruction of Diffraction Grating Imaging

In general, computational reconstruction of 3D scene from the multi-view images such as the elemental image array in integral imaging adopts the technique of reconstructing the sliced image for various target depth planes through the back-projection process [26,27,28]. However, in diffraction grating imaging, 3D computational reconstruction is performed using the characteristic that the spatial period of DIA is constant at the object’s depth. The computational reconstruction results of diffraction grating imaging at the object’s depth $z_{o}$ can be given by [13]
(7)$$R\left(x_{D}\right){|}_{z_{o}}=g\left(x_{D}\right)*\sum_{k=-1}^{1}\delta \left(x_{o}-kX_{z_{o}}\right),$$
where ∗ denotes convolution, $g(x_{D})$ is the intensity of DIA from Equation (Equation 6). The latter part of Equation (Equation 7) is a δ-function array with the spatial period, *X*, at the object’s depth, $z_{o}$, for computational reconstruction.

Figure 2 shows the result of computational reconstruction at the spatial period of the δ-function array in the convolution process of DIA and the δ-function array as a supplementary explanation for Equation (Equation 7). Figure 2a is the result when the δ-function array has a spatial period at the object’s depth, and Figure 2b is the result when they do not match each other. Each function in Equation (Equation 7) is written at the bottom of Figure 2b.

## 3. Photon Counting Method

3D objects under photon-starved conditions may not be visualized by conventional imaging system due to the lack of photons. Since photons occur rarely in unit time and space [25], a photon detecting method is required to visualize 3D objects under these conditions. Photon counting imaging can detect photons under photon-starved conditions as shown in Figure 3. However, physical photon detector is not cost-effective. Poisson distribution can be applied to detect photons because its characteristic is similar to the photon occurrence. Therefore, in this paper, we use computational photon counting method [15,16,17,18,19,20,21,22,23,24].

Photon counting imaging can record the images of objects under photon-starved conditions by detecting photons emitted from objects. In particular, computational photon counting method is more cost-effective than physical photon detector. It can be modelled by Poisson random process to generate photon counting images from the scene under photon-starved conditions. Figure 4 illustrates the procedure of the computational photon counting method. We assume that the scene under photon-starved conditions is the normalized irradiance which has the unit energy as shown in Figure 4.

Thus, using photon counting model such as Poisson distribution with the expected number of photons $N_{p}$, photon counting image can be generated. For convenience, we consider only 1D data. Computational photon counting model can be expressed as [15]
(8)$$\lambda_{x}=\frac{I_{x}}{\sum_{x}I_{x}},$$
(9)$$C_{x}|\lambda_{x}∼Poisson\left(N_{p}\lambda_{x}\right),$$
where $I_{x}$ is the original image, $\lambda_{x}$ is the normalized irradiance of the original image, $N_{p}$ is the expected number of photons from the normalized irradiance, and $C_{x}$ is the photon counting image by computational photon counting model.

To visualize or estimate the images, computational photon counting reconstruction, which uses multiple 2D photon counting images and advanced statistical estimation methods such as maximum likelihood estimation (MLE) or Bayesian estimation, can be applied. Since photon counting method can record multiple 2D images, the likelihood function can be constructed as [16]
(10)$$L\left(N_{p}\lambda_{k}|C_{k}\right)=\prod_{k=1}^{K}\frac{{\left(N_{p}\lambda_{k}\right)}^{C_{k}}e^{-N_{p}\lambda_{k}}}{C_{k}!},$$
(11)$$l\left(N_{p}\lambda_{k}|C_{k}\right)=\ln\left[L\left(N_{p}\lambda_{k}|C_{k}\right)\right]\propto \sum_{k=1}^{K}C_{k}\ln\left(N_{p}\lambda_{k}\right)-\sum_{k=1}^{K}N_{p}\lambda_{k},$$
where L(),l() are the likelihood and log likelihood functions, $\lambda_{k},C_{k}$ are the normalized irradiance and photon counting image of the kth recorded image, and *K* is the total number of the recorded images, respectively. We can find the optimum $\lambda_{k}$ for maximizing the likelihood function by MLE [16]
(12)$$\frac{\partial l(N_{p}\lambda_{k}|C_{k})}{\partial \lambda_{k}}=\frac{C_{k}}{\lambda_{k}}-N_{p}=0,$$
(13)$$∴{\hat{\lambda }}_{k}=\frac{C_{k}}{N_{p}}.$$

However, the estimated image by Equations (10)–(13) may have the low visual quality under severely photon-starved conditions since over ±1st order diffraction images have the low intensity. Therefore, to estimate the more accurate images, Bayesian estimation can be utilized. It can estimate the more accurate image because it uses the prior information of the original image. In general, since the pixel intensity of the image has a range [0,255], it is assumed that the prior information follows Gamma distribution as [16,18]
(14)$$\pi \left(\lambda_{k}\right)=\frac{\beta_{k}^{\alpha_{k}}}{\Gamma (\alpha_{k})}\lambda_{k}^{\alpha_{k}-1}e^{-\beta_{k}\lambda_{k}},\;\lambda_{k}>0,$$
(15)$$\mu_{k}=\frac{1}{N_{x}}\sum_{x}\lambda_{k}\left(x\right),\;\sigma_{k}^{2}=E{\left[\lambda_{k}-\mu_{k}\right]}^{2},$$
(16)$$\alpha_{k}=\frac{\mu_{k}^{2}}{\sigma_{k}^{2}},\;\beta_{k}=\frac{\mu_{k}}{\sigma_{k}^{2}},$$
where $\pi (\lambda_{k})$ is the statistical distribution (i.e., Gamma distribution) of the prior information, $\mu_{k},\sigma_{k}^{2}$ are the mean and variance of the original image, $N_{x}$ is the total number of pixels for the diffraction image, *E* is the expectation operator, and $\alpha_{k},\beta_{k}$ are the statistical parameters of Gamma distribution found by Equation (Equation 16), respectively. Therefore, by multiplying Equations (10) and (14), the posterior distribution can be obtained as a new Gamma distribution. To estimate the more accurate image, maximizing the posterior distribution is required as [16,18]
(17)$${\tilde{\lambda }}_{k}=\frac{C_{k}+\alpha_{k}}{N_{p}\left(1+\beta_{k}\right)},\;C_{k}>0.$$

Finally, the estimated image by Bayesian approach can be obtained. Figure 5 shows the examples of MLE and Bayesian estimation. It is noticed that Bayesian result can reduce more background noise than MLE result. However, it still has the low visual quality because the uncertainty of the detected photons may decrease the visual quality of the photon counting image. To improve the visual quality, in this paper, we propose a new photon counting model which uses multiple observation of photons. It is expressed in Section 4.

## 4. 3D Reconstruction under Photon-Starved Conditions Using Lensless 3D Imaging

In lensless 3D imaging such as diffraction grating imaging, the intensity of the recorded diffraction images may be low because of its low light efficiency. In particular, ±1st order diffraction images may have very low intensity under photon-starved conditions. Thus, conventional 3D image visualization method may not obtain 3D images with the sufficient light intensity. To solve this problem, photon counting method can be utilized. However, under severely photon-starved conditions, it may not generate the photon counting images with the sufficient photons. Therefore, a new photon counting technique is required. In this paper, we propose a photon counting model based on multiple observation of photon counting images.

Uncertainty of the detected photons may decrease the visual quality of the reconstructed 3D images by lensless 3D imaging. This uncertainty may be remedied by multiple observation of photon counting images which is called as *N* observation photon counting method because the samples of photon counting images can increase. When the single photon counting image is observed multiple times, the likelihood function of the single photon counting image can be constructed by
(18)$$L\left(N_{p}\lambda_{k}|C_{k,n}\right)=\prod_{n=1}^{N}\frac{{\left(N_{p}\lambda_{k}\right)}^{C_{k,n}}e^{-N_{p}\lambda_{k}}}{C_{k,n}!},$$
(19)$$l\left(N_{p}\lambda_{k}|C_{k,n}\right)=\ln L(N_{p}\lambda_{k}|C_{k,n})\propto \sum_{n=1}^{N}C_{k,n}\ln N_{p}\lambda_{k}-\sum_{n=1}^{N}N_{p}\lambda_{k},$$
where $C_{k,n}$ is the nth observation of kth photon counting image and *N* is the total number of observations. By maximizing the log likelihood function with the estimated parameter $\lambda_{k}$ (i.e., MLE), the estimated image can be obtained by
(20)$$\frac{\partial l(N_{p}\lambda_{k}|C_{k,n})}{\partial \lambda_{k}}=\frac{1}{\lambda_{k}}\sum_{n=1}^{N}C_{k,n}-N\times N_{p}=0,$$
(21)$$∴{\hat{\lambda }}_{kN}=\frac{1}{N}\sum_{n=1}^{N}\frac{C_{k,n}}{N_{p}},\;C_{k,n}>0.$$

In *N* observation photon counting method, the estimated image has more accurate intensity than the single observed photon counting image. However, MLE with *N* observations may not estimate the accurate image under severely photon-starved conditions since it uses Uniform distribution as the prior information. Therefore, the more accurate image can be estimated by using Bayesian estimation where the prior information follows Gamma distribution. Posterior distribution can be written as
(22)$$\pi \left(\lambda_{k}|C_{k,n}\right)\propto \prod_{n=1}^{N}{\left(N_{p}\lambda_{k}\right)}^{C_{k,n}}e^{-N_{p}\lambda_{k}}\times \lambda_{k}^{\alpha_{k}-1}e^{-\beta_{k}\lambda_{k}}\propto \lambda_{k}^{\sum_{n=1}^{N}C_{k,n}+\alpha_{k}-1}e^{-\left(NN_{p}+\beta_{k}\right)\lambda_{k}},$$
(23)$$\pi \left(\lambda_{k}|C_{k,n}\right)∼\Gamma \left(\sum_{n=1}^{N}C_{k,n}+\alpha_{k},NN_{p}+\beta_{k}\right).$$

By maximizing the posterior distribution (i.e., maximum a posterior (MAP)), the estimated image can be obtained as
(24)$$∴{\tilde{\lambda }}_{kN}=\frac{\alpha_{k}+\sum_{n=1}^{N}C_{k,n}}{NN_{p}+\beta_{k}},\;C_{k,n}>0.$$

Figure 6a shows the single observation photon counting images with different expected photon ratios. The expected photon ratio can be calculated by $R_{p}=N_{p}/N_{x}$. They may not be recognized well. In contrast, the N=100 observation photon counting images by MLE and MAP with different expected photon ratios as shown in Figure 6b,c have the sufficient visual quality for recognition. Under severely photon-starved conditions (i.e., 1% photon ratio), it is noticed that the estimated image by MAP (see Figure 6c) is more accurate than the one by MLE (see Figure 6b). As the photon ratio increases, the estimation accuracy for both methods is the almost same because the samples of photons increase. Finally, 3D images of lensless 3D imaging can be obtained by using Equation (Equation 7). We will show the experimental results of 3D images in Section 5.

## 5. Experimental Results

Figure 7 illustrates the experimental setup and shows the diffraction images. The test objects were placed 100 mm away from a diffraction grating, and the diffraction grating was located at 400 mm from a camera as shown in Figure 7a. The size of the objects used in the experiment and the distance between them are illustrated in Figure 7b. The diffraction grating in this experiment is made of two transmissive amplitude diffraction gratings with a line density of 500 lines/mm that are available commercially through perpendicularly concatenating two diffraction gratings. The objects are illuminated using a laser diode with the optical power of 4.5 mW and the wavelength of 532 nm. A digital camera with a CMOS sensor of 36 (H) × 24 (V) mm and a pixel pitch of 5.95 μm is used to acquire the DIA as shown in Figure 7c. Each diffraction image of HKNU and Men objects has 561 (H) × 561 (V) pixels, respectively. The field of view of the diffraction grating imaging system used in the experiment is approximately 18 degrees and can be calculated through the system configuration as shown in the left side of Figure 7a.

In diffraction grating imaging, the limitation of the object’s image size is discussed in Equation (Equation 7) of Reference [12], and the object’s image size can be increased as the distance between the diffraction grating and the object increases. Diffraction grating imaging allows diffraction images to overlap each other when the size of an object exceeds a certain limit. In diffraction grating imaging, the maximum size of an object that overlaps between diffraction images is defined as effective object area (EOA) [12]. EOA can be calculated from the relations of the wavelength of the light source, the spatial resolution of the diffraction grating and the distance between the object and the diffraction grating, and it is given by
(25)$$\begin{array}{c} EOA=\left|z_{o}-d\right|\tan\left(\sin^{-1}\left(\frac{\lambda }{a}\right)\right) \end{array}$$

The EOA according to the distance of the object in the diffraction grating imaging system used in this experiment is shown in Figure 8.

Figure 9 and Figure 10 show the photon counting images generated from the −1st, 0th, +1st order diffraction images of HKNU and Men by the single observation, N=100 observation MLE, and N=100 observation MAP of photon counting method with 1% photon ratio (i.e., 0.01×561×561=3147 photons). It is noticed that photon counting images by N=100 observation of MLE and MAP have better visual quality than photon counting images by the single observation. In addition, the visual quality of the 0th order image by MAP is better than the one by MLE.

Using computational reconstruction of diffraction grating imaging as written in Equation (Equation 7), 3D images with various spatial periods at reconstruction depths can be obtained as shown in Figure 11. These reconstructed 3D images are used as the reference images for calculating the performance metric of our proposed method such as peak sidelobe ratio (PSR) [17]. Figure 12 and Figure 13 show the reconstructed 3D images of HKNU and Men with various spatial periods at reconstruction depths under photon-starved conditions by using the single observation, N=100 observation MLE, and N=100 observation MAP, respectively, where the photon ratio is 1%. As shown in Figure 12 and Figure 13, 3D images by N=100 observation of MLE and MAP have better visual quality than the single observation. In addition, their visual quality is the almost same as the reference images as shown in Figure 11. However, the difference of the visual quality between MLE and MAP may not be significant. Therefore, in this paper, we calculate the PSR values of the reconstructed 3D images by the single observation, N=100 observation MLE, and N=100 observation MAP via various spatial periods at reconstruction depths as shown in Figure 14 and Figure 15, respectively. PSR of the correlation peak is defined as the number of standard deviation by which the peak exceeds the mean value of the correlation surface. It can be calculated by [17]
(26)$$\begin{array}{c} PSR=\frac{\max\left[c(x)\right]-\mu_{c}}{\sigma_{c}} \end{array}$$
where $\mu_{c}$ is the mean of the correlation and $\sigma_{c}$ is the standard deviation of the correlation. The higher the PSR value is, the better the recognition performance obtained.

In Figure 14 and Figure 15, the highest peak PSR values at the locations of 3D objects. In particular, PSR results by *N* observation MLE and *N* observation MAP have approximately three times higher values than PSR results by the single observation with each spatial period at reconstruction depth of objects. In addition, the slope of PSR graph for MAP is steeper than the one for MLE. It means that the longitudinal resolution of MAP is better than MLE. On the other hand, PSR values by single observation photon counting method are relatively very low and the slope of PSR graph is gradual. Therefore, our proposed method, the lensless 3D imaging using diffraction grating and *N* observation photon counting method, can visualize 3D images and improve their visual quality under photon-starved conditions.

## 6. Conclusions

In this paper, we have proposed a lensless 3D imaging under photon-starved conditions using diffraction grating imaging and *N* observation photon counting method. In conventional lensless 3D imaging, it may be difficult to visualize 3D images under photon-starved conditions due to lack of photons. On the other hand, in our proposed method, using diffraction grating imaging and *N* observation photon counting method, this difficulty can be solved. Since our proposed method can record 3D information by the single shot and without lens under photon-starved conditions, it can obtain dynamic 3D images without diffraction and aberrations. In addition, our proposed method can obtain more accurate 3D images by using Bayesian estimation with *N* observation photon counting method. Therefore, since our proposed method can visualize 3D objects under photon-starved conditions by lensless imaging, we believe that it can be utilized to various applications such as 3D medical imaging under low light level environment, nigh vision for unmanned autonomous vehicle, security camera, and so on.

## Figures and Tables

**Figure 1 sensors-23-02336-f001:**
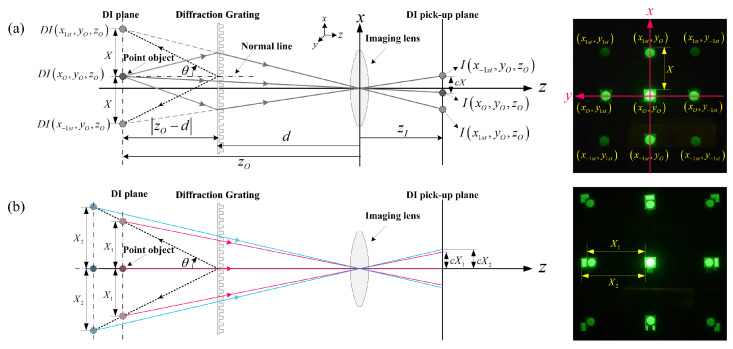
Geometric relations in diffraction grating imaging and examples of diffraction image array (DIA). (**a**) On the left, geometric relationship for point object, diffraction images (DIs), diffraction grating and imaging lens. On the right, it is an example of the DIA; (**b**) On the left, the spatial period of the diffraction image array according to the depth of the object. On the right, it is the example of the DIA.

**Figure 2 sensors-23-02336-f002:**
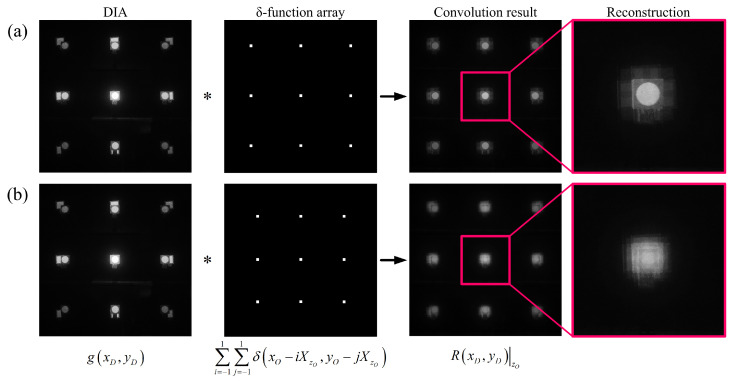
Computational reconstruction through convolution of DIA and δ-function arrays in diffraction grating imaging. (**a**) Reconstruction result when the spatial period of the δ-function array coincides with the spatial period at the object’s depth; (**b**) Reconstruction result when the spatial period at the object’s depth and that of the δ-function array do not match each other.

**Figure 3 sensors-23-02336-f003:**
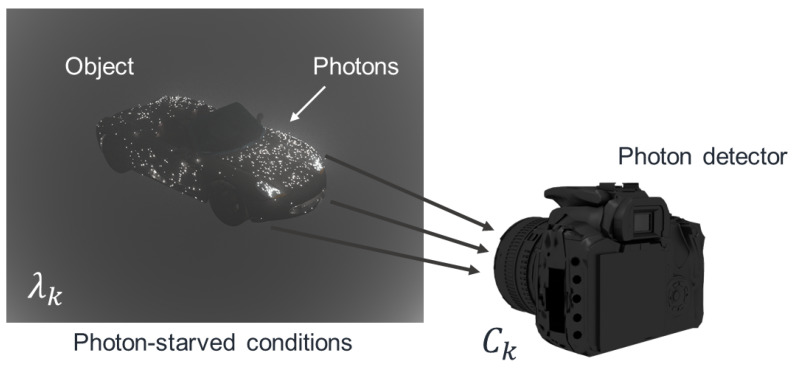
Physical photon counting detector.

**Figure 4 sensors-23-02336-f004:**
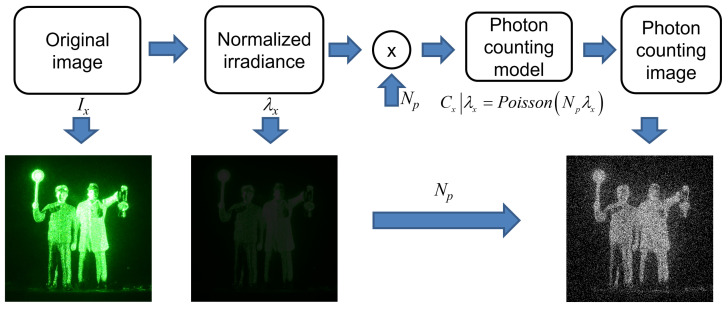
Procedure of computational photon counting model.

**Figure 5 sensors-23-02336-f005:**
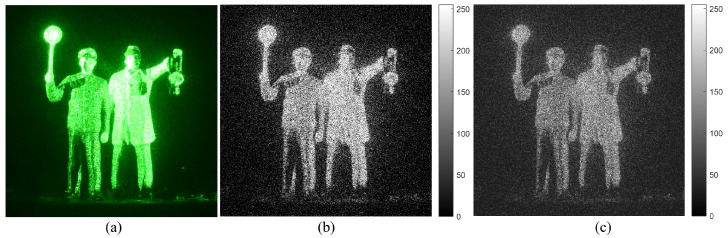
(**a**) Original image, (**b**) photon counting image by MLE, and (**c**) photon counting image by Bayesian estimation where 157,361 photons are extracted from the original image (**a**).

**Figure 6 sensors-23-02336-f006:**
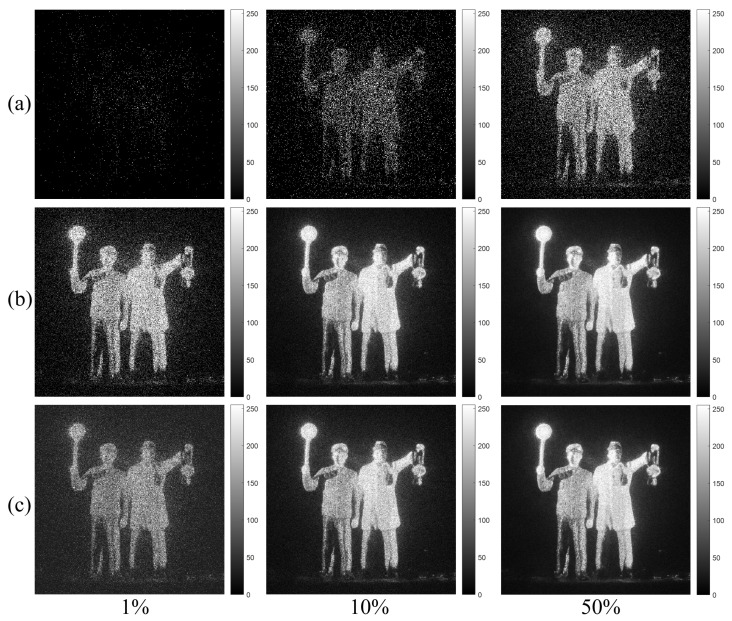
Estimated images with different expected photon ratios (1%, 10%, and 50%). (**a**) Single observation photon counting images, (**b**) N=100 observation photon counting imaging by MLE and (**c**) N=100 observation photon counting imaging by Bayesian estimation, respectively.

**Figure 7 sensors-23-02336-f007:**
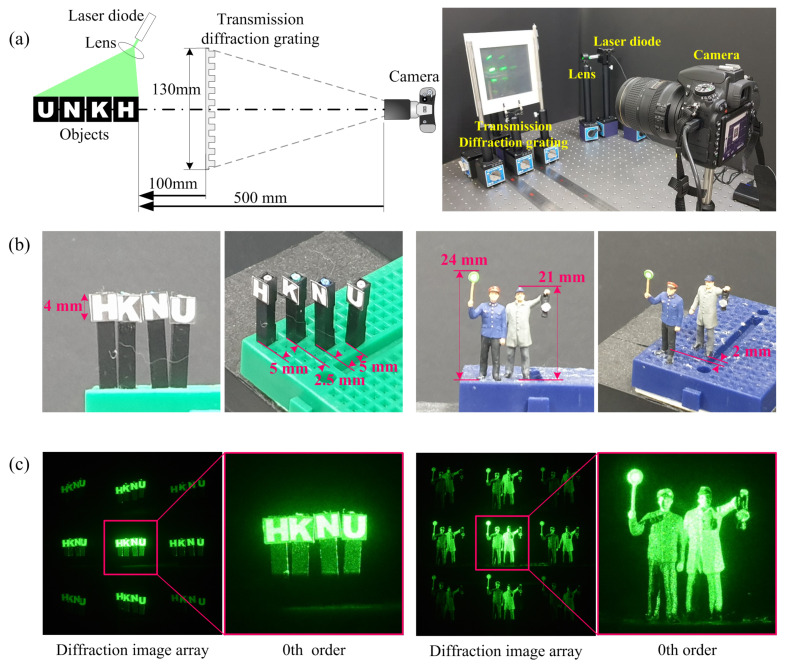
Optical experiment setup to acquire DIA. (**a**) Configuration of optical experiment and (**b**) The size of the objects used in the experiment and the distance between them and (**c**) diffraction image arrays (DIA) and the enlarged images of their 0th order diffraction images.

**Figure 8 sensors-23-02336-f008:**
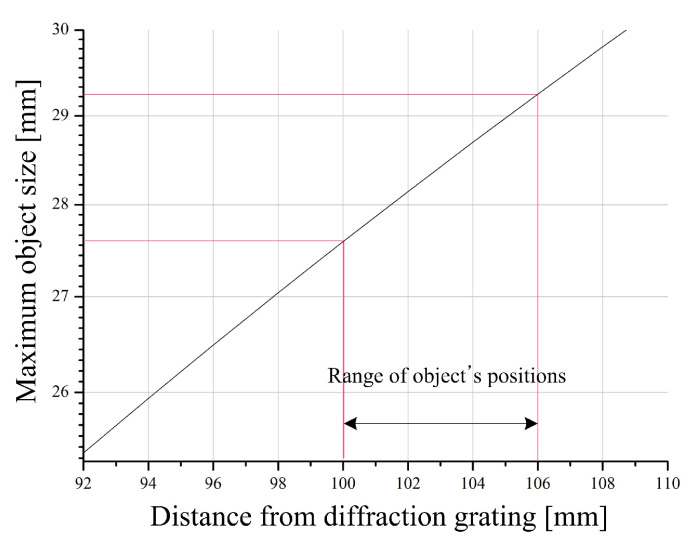
EOA for the distance between the diffraction grating and the object in this diffraction grating imaging system.

**Figure 9 sensors-23-02336-f009:**
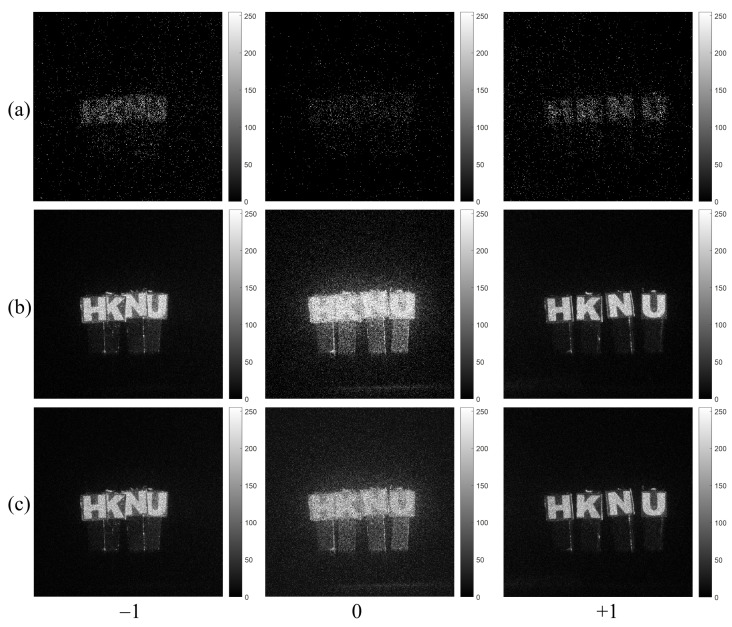
Diffraction images of HKNU objects by photon counting imaging with 1% photon ratio by (**a**) Single observation, (**b**) N=100 observations of MLE, and (**c**) N=100 observations of MAP, respectively.

**Figure 10 sensors-23-02336-f010:**
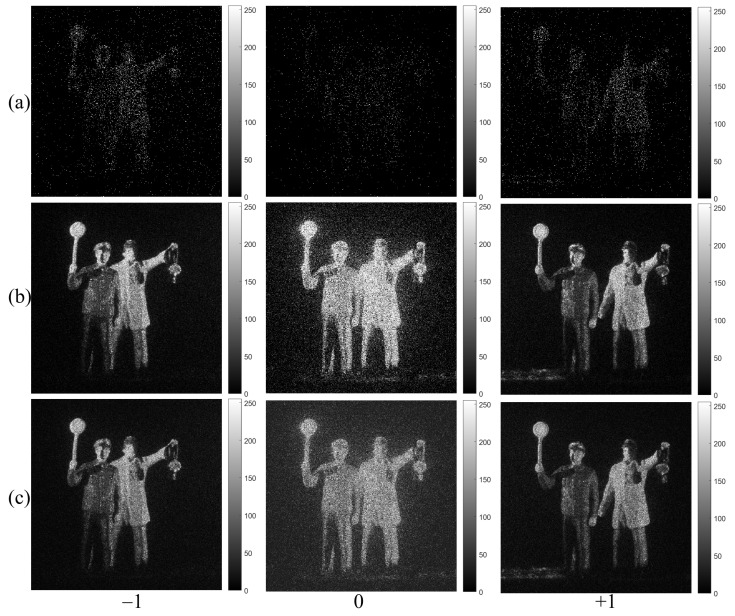
Diffraction images of Men objects by photon counting imaging with 1% photon ratio by (**a**) Single observation, (**b**) N=100 observations of MLE, and (**c**) N=100 observations of MAP, respectively.

**Figure 11 sensors-23-02336-f011:**
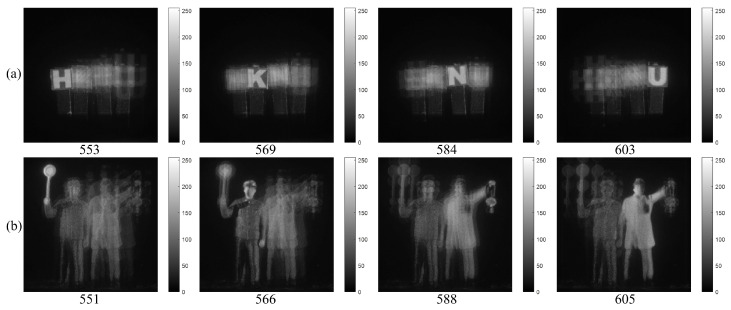
3D images under normal illumination with various spatial periods at reconstruction depths by the original diffraction images of (**a**) HKNU objects and (**b**) Men objects, respectively.

**Figure 12 sensors-23-02336-f012:**
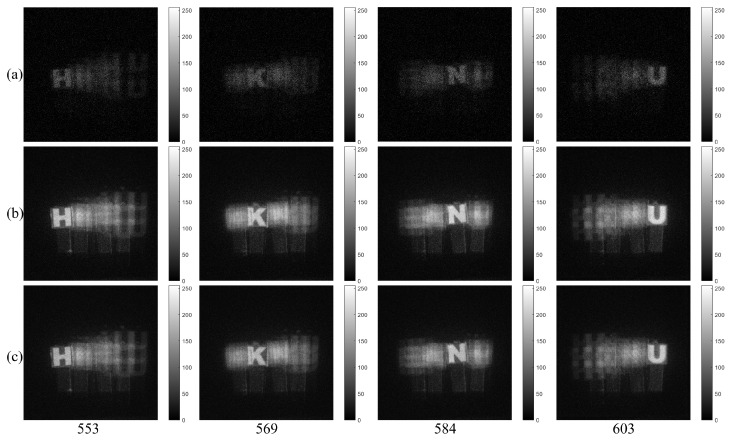
3D images under photon-starved conditions of HKNU objects with 1% photon ratio and various spatial periods at reconstruction depths by (**a**) the single observation photon counting images (**b**) N=100 observation photon counting images by MLE and (**c**) N=100 observation photon counting images by MAP, respectively.

**Figure 13 sensors-23-02336-f013:**
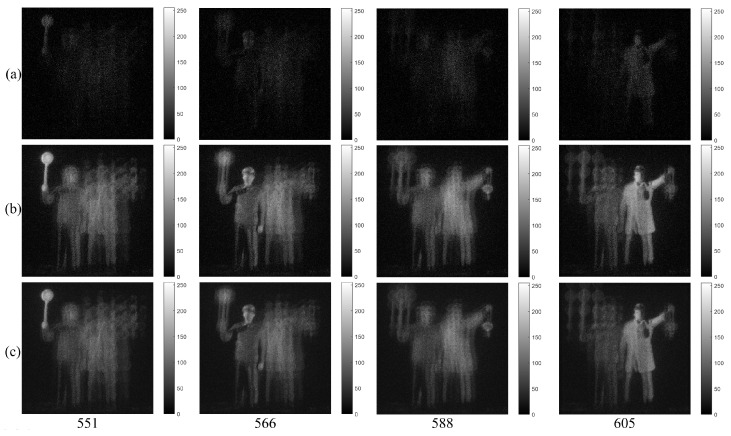
3D images under photon-starved conditions of Men objects with 1% photon ratio and various spatial periods at reconstruction depths by (**a**) the single observation photon counting images (**b**) N=100 observation photon counting images by MLE and (**c**) N=100 observation photon counting images by MAP, respectively.

**Figure 14 sensors-23-02336-f014:**
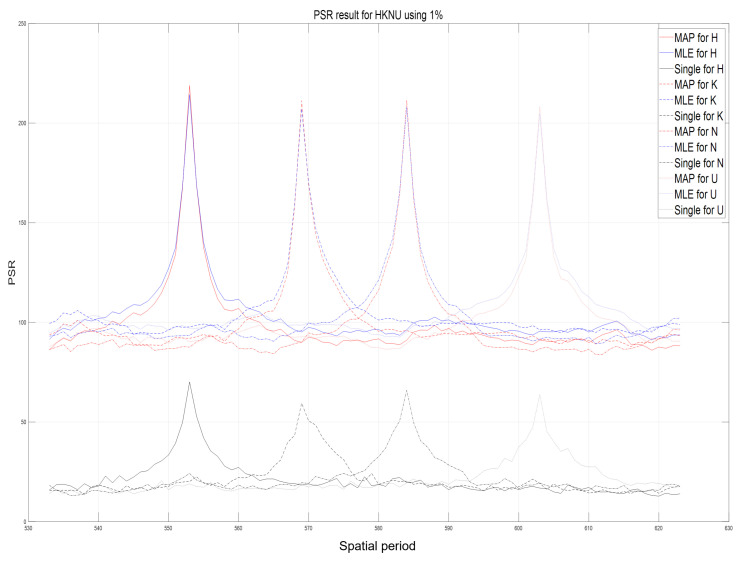
Peak sidelobe ratio (PSR) results of HKNU objects via various spatial periods at reconstruction depths by single observation photon counting method, N=100 observation MLE, and N=100 observation MAP with 1% photon ratio.

**Figure 15 sensors-23-02336-f015:**
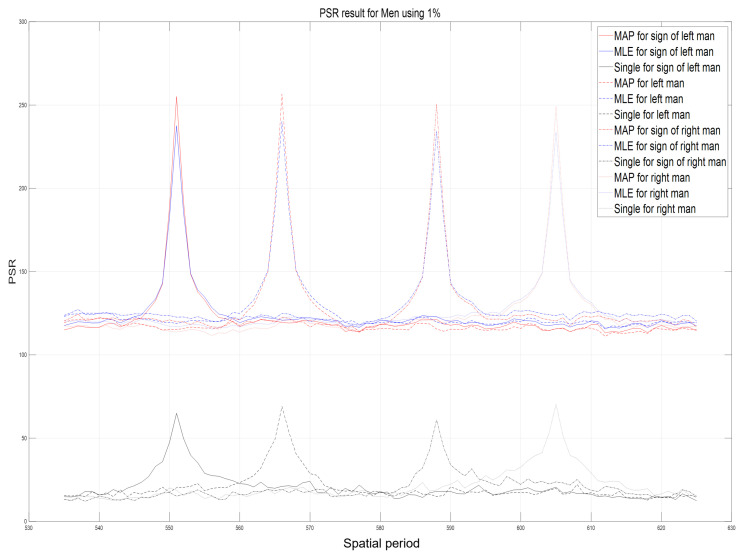
Peak sidelobe ratio (PSR) results of Men objects via various spatial periods at reconstruction depths by single observation photon counting method, N=100 observation MLE, and N=100 observation MAP with 1% photon ratio.

## Data Availability

Not applicable.

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
