# Peer review of "Lensless Three-Dimensional Imaging under Photon-Starved Conditions"

_sensors, 2023, doi:10.3390/s23042336_

Round 1

Reviewer 1 Report

The manuscript propose a lens less three dimensional imaging under photon-starved conditions using diffraction grating and computational photon counting method. The approach utilizes multiple observation photon counting method with advanced statistical estimation to improve the visual quality of 3D images under severely photon starved conditions. Some suggestions regarding the current manuscript are as follows.

1.     The manuscript need to provide more details of the specification of the object used in the experimental demonstration (size, material, depth, transmitting or reflecting type, etc.).

2.     Is there any imaging size limitation of the object that can be imaged using the proposed system? Also, a discussion on the field of view of the proposed system may demonstrate the applicability of the system.

3.     The figures with imaging results requires the respective scale bars and corresponding intensity maps.

4.     Is this approach can be directly applied to any real valued objects? The manuscript needs to provide more discussions by highlighting the practical applications of the system.

Reviewer 2 Report

In this work, authors presented an algorithm for reconstructing 3D images based on the computational photon counting method. It was called by authors, the N observation photon counting method, which consisted in taking multiple observations of photon counting method. Furthermore, according to authors by using the proposed method the visual quality of reconstructed 3D images of objects, under starving photon conditions, can be considerably enhanced. In general, de main idea is interesting and the manuscript is well written, there are only some points that requires some clarifications. Based on these considerations, I would like to suggest some moderate changes in the manuscript prior to be considered for publication in the journal.  

Some points that must be addressed are:

1-      In figure 7a it is shown the sketch of the experimental setup, however, in figure 7b and within the text it is mentioned that a laser (green) is used. Therefore I would like to suggest to include the laser in the setup drawing of Fig. 7a. Moreover, in Fig. 7b it can be appreciated that in front of the laser is a component, please mention if is a lens/collimator, and include it the Fig. 7a. Additionally, within the Fig. 7a and within the manuscript text please specify if the objects (letters and men) are placed at certain angle/position with respect of the laser and the refraction grating, because it is not clear, how these are positioned.

2-      I would like to recommend, for clarity purposes, to explain with more detail the peak sidelobe ratio (PSR) concept and how it is calculated. Moreover, provide a deeper explanation of how the performance metric of your system can be determined based on the PSR results shown in Figs. 13 and 14.

Round 2

Reviewer 2 Report

I consider that authors have attended the comments provided in the first revision round, and therefore I would like to recommend accept in its present form.